# The CXCL12 Crossroads in Cancer Stem Cells and Their Niche

**DOI:** 10.3390/cancers13030469

**Published:** 2021-01-26

**Authors:** Juan Carlos López-Gil, Laura Martin-Hijano, Patrick C. Hermann, Bruno Sainz

**Affiliations:** 1Department of Cancer Biology, Instituto de Investigaciones Biomédicas “Alberto Sols” (IIBM), CSIC-UAM, 28029 Madrid, Spain; jclopez@iib.uam.es (J.C.L.-G.); lmartin@iib.uam.es (L.M.-H.); 2Department of Biochemistry, Universidad Autónoma de Madrid (UAM), 28029 Madrid, Spain; 3Chronic Diseases and Cancer, Area 3-Instituto Ramon y Cajal de Investigación Sanitaria (IRYCIS), 28029 Madrid, Spain; 4Department of Internal Medicine I, Ulm University, 89081 Ulm, Germany

**Keywords:** cancer stem cells, CSC niche, CXCL12, CXCR4, CXCR4 inhibitors

## Abstract

**Simple Summary:**

CXCL12 and its receptors have been extensively studied in cancer, including their influence on cancer stem cells (CSCs) and their niche. This intensive research has led to a better understanding of the crosstalk between CXCL12 and CSCs, which has aided in designing several drugs that are currently being tested in clinical trials. However, a comprehensive review has not been published to date. The aim of this review is to provide an overview on how CXCL12 axes are involved in the regulation and maintenance of CSCs, their presence and influence at different cellular levels within the CSC niche, and the current state-of-the-art of therapeutic approaches aimed to target the CXCL12 crossroads.

**Abstract:**

Cancer stem cells (CSCs) are defined as a subpopulation of “stem”-like cells within the tumor with unique characteristics that allow them to maintain tumor growth, escape standard anti-tumor therapies and drive subsequent repopulation of the tumor. This is the result of their intrinsic “stem”-like features and the strong driving influence of the CSC niche, a subcompartment within the tumor microenvironment that includes a diverse group of cells focused on maintaining and supporting the CSC. CXCL12 is a chemokine that plays a crucial role in hematopoietic stem cell support and has been extensively reported to be involved in several cancer-related processes. In this review, we will provide the latest evidence about the interactions between CSC niche-derived CXCL12 and its receptors—CXCR4 and CXCR7—present on CSC populations across different tumor entities. The interactions facilitated by CXCL12/CXCR4/CXCR7 axes seem to be strongly linked to CSC “stem”-like features, tumor progression, and metastasis promotion. Altogether, this suggests a role for CXCL12 and its receptors in the maintenance of CSCs and the components of their niche. Moreover, we will also provide an update of the therapeutic options being currently tested to disrupt the CXCL12 axes in order to target, directly or indirectly, the CSC subpopulation.

## 1. Introduction

### 1.1. The Cancer Stem Cell Model

Although several theories have been proposed to explain the origin of tumors, the cancer stem cell (CSC) concept has become a well-accepted model that accurately explains tumor origin, heterogeneity and hierarchy. The concept of a population of disease-producing “altered cells” with the ability to generate clones of themselves was first described by Rudolf Virchow in 1885 in a manifest defending animal scientific research [1]. Since then, the concept of CSCs has evolved significantly, with John Dick and colleagues first experimentally proving the existence of CSCs in acute myeloid leukemia (AML) in 1994 [2]. Since then, CSCs have been identified in a multitude of hematological and solid malignancies such as colorectal [3], lung [4], and brain tumors [5], among others. While their existence has been extensively debated, it is now accepted that CSCs constitute a population of cells with self-renewal, chemoresistant, and metastasis- and tumor-initiating capacities. Today, CSC research largely focuses on trying to unravel the origin of these cells, the mechanisms that underlie their inherent plasticity, their precise role in metastasis and chemoresistance, and their suitability as a therapeutic target, the latter relying on the ability to identify, isolate, and target them based on markers or properties that set them apart from other cells (including normal stem cells).

One of the characteristics associated with CSCs is plasticity. Different opinions about the CSC model have led researchers to rethink the intrinsic abilities of this subset of cells with respect to differentiation and dedifferentiation. The classic hypothesis regarding CSC division is in accordance with the general model of stem cell division, in which a stem cell is able to undergo branch-like division. On the one hand, symmetric division results in two stem cells, ensuring that the CSC pool is never depleted. On the other hand, asymmetric division gives rise to a stem cell and a non-stem cell, the latter of which is usually referred to as progenitor or transient-amplifying daughter cell. While these maintain certain stem-like properties, it is generally believed that their pluripotency is decreased compared to the parental CSC [6]. As these progenitor/daughter cells divide, they give rise to more differentiated “less CSC-like” cells; however, the process of differentiation is no longer considered to be a path of no return. Similar to what has been shown for normal tissues such as the intestinal crypt [7], cancer cells in intermediate states of differentiation are believed to be plastic, i.e., to be capable of reverting to a more dedifferentiated CSC-like state if the correct signals are present. This cellular reprogramming could be the result of signals exchanged between non-CSCs and the tumor microenvironment (TME) [8]. A review regarding the plasticity of CSCs in pancreatic cancer by Hermann and Sainz present data to support both the state and the entity hypotheses, based largely on two studies describing how Lgr5^+^ colorectal CSCs can originate from a Lgr5^−^ subset of non-CSC cells, reinforcing the hypothesis that cancer stemness is more a state than an entity [9,10,11].

The role of plasticity is also highly relevant during metastasis. Epithelial-to-mesenchymal transition (EMT) is currently the accepted theory to explain how a cancer cell loses its adhesion to the extracellular matrix (ECM) and to other tumor cells in order to invade the surrounding tissue and disseminate after entering into circulation. Circulating tumor cells then find a receptive niche to seed and, through mesenchymal-to-epithelial transition (MET), the reversal of EMT, revert to their epithelial state and form secondary tumors [12]. This entire process requires immune adaptation to allow tumor cells to evade cytotoxic immune cells while in circulation and to recruit protumoral immune cells once at the metastatic site. Nowadays, EMT and MET are considered to be a continuous and intertwined spectrum of molecular and phenotypic changes, in which the epithelial and mesenchymal phenotypes are not permanent but transient, and characterized by fully polarized or hybrid EMT/MET states, which can be identified by transcriptional profiling and analysis of surface markers such as vimentin or E-cadherin expression [13,14,15]. A comprehensive consensus, the result of a two-year work to clarify the guidelines for EMT transition, has been recently published [16]. At the CSC level, in 2008, Mani et al. established a link between EMT and stemness. Specifically, they induced EMT in human mammary epithelial cells, which induced stem-like features in these cells [17]. This work set the precedent for others studies, which, with a higher grade of refinement, would show that the highest level of cancer stemness corresponds to an epithelial/mesenchymal hybrid state that retains both mesenchymal and epithelial traits [13], explaining (at least in part) the level of cellular plasticity inherent to these cells and validating the role of EMT in CSC stemness.

Although CSC biology has been studied intensively, their identification and isolation has been one of the major challenges in the field and at the same time the main objective of the critics of the CSC model. However, regardless of the methods used, sets of well-defined markers exist, have been described and published, and serve as a way to isolate CSC populations from various tumor entities [18]. While many markers or marker combinations are specific for certain tumor entities, others are commonly expressed in different tumor types, and are linked to functional properties. In addition to so-called CSC markers, CSCs can be differentiated from non-CSCs at the genetic, epigenetic, and transcriptional level, as evidenced by the reactivation of embryonic signaling pathways such as Wnt, Nodal, or Notch [19,20,21], or by the overexpression of genes related to pluripotency and cell reprogramming such as *NANOG*, *SOX2*, *OCT4*, or *KLF4*, commonly known as the “Yamanaka factors” [22].

At the level of CSC markers, CD44, CD24, aldehyde dehydrogenase 1 (ALDH1), CD133, and CXCR4, the latter being a core part of this review, represent widely used markers to identify CSCs (although none of these are exclusive to CSCs or of one CSC population). Moreover, these markers are very useful to isolate CSCs by sorting specific and functionally distinct subpopulations via fluorescence-activated cell sorting (FACS). CD44 is a non-kinase transmembrane receptor that contributes to adhesion between cells and matrix proteins, and participates in cell growth and migration [23]. High levels of CD44 expression in pancreatic ductal adenocarcinoma (PDAC) and colorectal cancer (CRC) correlate with poor prognosis, and expression of the variant CD44v6 indicates a higher metastatic potential. While the standard isoform is the most commonly expressed on PDAC CSCs (PaCSCs), the CD44v6 variant is more prevalent on CRC CSCs [24,25]. In some cancer types with high expression of CD44, there is a concomitant reduction in CD24, a cell surface protein that is related to cell adhesion [26]. Interestingly, in breast cancer this co-regulation seems to confer stemness, as CD44^high^/CD24^low^ cells exhibit a higher tumorigenic potential in immunocompromised mice compared to CD44^low^/CD24^high^ cells. Furthermore, ALDH1 expression is positively associated with these markers. This enzyme plays a fundamental role in stem cells due to its differentiation- and detoxification-related functions, and a higher activity of ALDH1 along with CD44^high^/CD24^low^ expression correlates with tumor progression [27]. Last, CD133 is a transmembrane protein, which has been widely described to be expressed on the surface of CSCs in many different tumor entities. CD133 also correlates negatively with disease-free survival and positively with cell migration and metastasis [26]. Its expression in pancreatic and prostate cancer, hepatocellular carcinoma (HCC), and glioma tumor cells has served as a way to track and study the function of CSCs and their interaction with the TME [28,29,30].

As important as the intrinsic factors that define CSCs are the extrinsic surrounding factors (i.e., the TME) that can also govern the CSC’s phenotype and cellular fate. Among all the factors and cells that make up the TME, there are some specific components that specifically interact with CSCs and may have great importance in disease progression and therapeutic response. Below we will highlight a few of these important TME players.

### 1.2. The Cancer Stem Cell Niche

In 2011, with the update to “Hallmarks of Cancer”, the concept of the TME was defined as “the non-cancer cells that also make up a large part of the tumor and all the soluble and secreted factors which mediate the interactions between these cells” [31]. Along these lines, the CSC niche (as part of the TME), also plays an important role in driving a tumor’s progression and its adaptation to hostile conditions. The CSC niche is thought to be composed of different cell types (e.g., differentiated cancer cells and stromal cells) and structures (e.g., blood and lymphatic vessels) and is surrounded by an extracellular matrix (ECM) [32]. The ECM consists of different types of molecules, which together create a supportive 3D structure that regulates cell migration, growth, and differentiation among other processes, and its constant remodeling forms part of that regulation [33]. Aberrant remodeling of the ECM is a hallmark of cancer and the expression of specific proteins such as periostin and tenascin in breast CSCs, for example, are required to start the metastatic process [34,35]. The role of the ECM in CSC regulation and chemoresistance has been thoroughly reviewed by Brown et al. [36]. Moreover, the CSC niche influences cell plasticity, conditioning cells to acquire a CSC phenotype. When the Wnt signaling pathway is overactivated, in large part due to factors like hepatocyte growth factor (HGF) secreted by fibroblasts, CRC cells acquire a higher tumorigenic and self-renewal potential [19]. Furthermore, the proinflammatory state of the niche positively modulates the NF-κB cascade, which can act on CSCs and non-CSCs alike. Regarding the latter, NF-κB activation can induce the de-differentiation [37] of non-stem cells towards tumor-initiating cells in an intestinal tumorigenesis mouse model.

In addition to the ECM, stromal cells are also crucial players in the CSC niche, and have been shown to regulate CSCs and each other via a crosstalk orchestrated by CSCs (Figure 1). Among all the different cell types present in the CSC niche, mesenchymal stem cells (MSCs), cancer-associated fibroblasts (CAFs), immune cells, and vascular endothelial cells constitute the most numerous cells [32,38,39]. In the following sections, we will review the main characteristics of these cell types and how they influence the status or activation of CSCs.

#### 1.2.1. Mesenchymal Stem Cells

The term “mesenchymal stem cell” has been widely discussed in other reviews [40,41], being used to describe those cells isolated from the bone marrow (BM) and positive for the markers CD105, CD73, and CD90 and negative for CD45, CD34, CD14, CD11b, CD19, and HLA-DR in vitro. In addition to the expression of the aforementioned markers, a defining feature of these cells is that they must be able to generate three lineages of stromal cells: chondrocytes, adipocytes, and osteoblasts [42]. MSCs are usually located in the bone marrow and in adipose tissue and have the potential to migrate to different locations in a process called “homing”, to participate in wound healing and tissue repair processes. The path followed by these cells is dictated by a gradient of cytokines and soluble factors secreted from the homing site. The chemotactic/homing process is also mediated by chemokine receptors present on the surface of MSCs, e.g., CXCR4 [43] and Potential for Homing.

Several studies have shown that cancer-related tissue damage and repair is very similar to that of other forms of injury [44,45]. Thus, MSCs are recruited to the TME much in the same way as they are to other damaged tissues to provide cellular signals and structural support, the latter of which is achieved by their differentiation into CAFs [46]. Once in the TME, MSCs modulate the deposition of some ECM components such as laminin, fibronectin, and collagen, and also release extracellular vesicles (EVs), which may modulate the total amount of ATP in the TME. Thus, MSC-derived EVs would enhance metabolic activity and proliferation of cancer cells [47]. Furthermore, MSC-derived EVs can also directly affect the fate of CSCs as EVs can contain TGF-β, IL-6, and other factors that modulate important stem cell-related signaling pathways like STAT3 or WNT [48]. Cancer cell-derived EVs also contribute to this crosstalk by inducing the polarization of stroma cells into a protumoral state [49]. In a different study, MSC-derived EVs were even shown to induce vasculature growth [50].

The polarization of MSCs has not been extensively studied in cancer. However, Waterman et al. have described two different populations of MSCs (anti- and proinflammatory), polarized through activation of toll-like receptors (TLR), where the anti-inflammatory MSC subpopulation is believed to increase tumor progression and migration by impeding T cell activation [51]. In contrast, Guilloton et al. have reported that MSCs from patients with follicular lymphoma displayed a proinflammatory gene expression profile compared to those obtained from healthy donors. These cancer-associated MSCs were capable of driving monocyte differentiation towards a protumoral phenotype, very similar to what has been observed for tumor-associated macrophages (TAMs) [52]. Apart from immune modulation, MSCs can also contribute to increased tumor chemoresistance. In addition to the innate chemoresistance attributed to CSCs, which express ATP-binding cassette transporters intracellularly and on their cell membrane [13], MSCs have the ability to indirectly interfere with chemotherapeutic agents. For example, a study published in 2011 showed that injection of MSCs into mice bearing subcutaneous breast, colorectal, and lung tumors correlated with enhanced chemoresistance to cisplatin. It was proposed that cisplatin induced the MSC population to secrete polyunsaturated fatty acids, which in turn blocked platinum-derived therapies, thus providing an additional chemoprotective property to the CSC niche [53].

As mentioned above, MSCs can directly regulate CSC fate via several mechanisms, which can differ based on the tumor entity. It was reported by Coffman et al. that MSCs can acquire a protumoral phenotype that promotes stemness in ovarian cancer via a signaling loop that involves tumor-derived Hedgehog and MSC-derived BMP4 [54]. Increased BMP4 levels are characteristic for normal MSCs as compared to protumoral MSCs. In the aforementioned study, this loop drove the enrichment of ALDH1^+^ CSCs both in vivo and in vitro, which again correlated with higher resistance to cisplatin [54]. In breast cancer, MSCs potentiate stemness by inducing microRNA-199a and microRNA-214 expression in cancer cells, which have been shown to repress Forkhead box protein P2 (FOXP2) expression and thus enhance mammary CSC features. Interestingly, CSCs expressing both microRNAs were more metastatic when injected into immunocompromised mice. Moreover, this microRNA signature is typical of highly aggressive breast cancers with a poor outcome [55]. Another study in breast cancer showed that co-culturing MSCs with tumor cells enriches for the CD44^+^/CD24^−^ CSC population. These MSCs maintained their osteogenic and adipogenic potential and increased the expression of cytokines such as *CXCL5*, *CXCL6*, *IL-6*, or *IL-8* both in MSCs and CSCs. MSC homing has also been observed in mouse models, where MSC migration was mediated through IL-6 secreted by CSCs [56]. MSC-conditioned medium was also shown to contain IL-6 and IL-8 when they were co-cultured with CRC cells, activating NF-κB and AMPK/mTOR signaling and inducing the expression of *OCT4*, *C-MYC*, and *SOX2* in CRC cells [57]. In a separate study, Li et al. described that the communication between CRC cells and MSCs in co-culture experiments not only resulted in the activation of these same signaling pathways observed by Wu et al. [57], but CRC cells also secreted IL-1, which activated the expression of prostaglandin E2, increasing the ALDH1^high^ CRC CSC population in vitro. In agreement with the concept of CSC plasticity, this study also showed how the ALDH1^high^ CSC subpopulation was able to return to its previous phenotype (i.e., ALDH1^low^) after the removal of MSCs from the culture, highlighting just how important niche signals are for maintaining (or activating) the CSC population [58]. In hormone-dependent tumors like prostate cancer, MSCs from the bone marrow downregulated the expression of the androgen receptor in tumor cells and increased the CSC population. Interestingly, when the infiltration of MSCs into the tumor was inhibited, the metastatic ability of tumor cells was decreased concomitantly with the CSC population [59]. While the role of MSCs in the immunomodulation of the CSC niche has not been fully elucidated to date, it is well known that MSCs are involved in angiogenesis, as they produce factors that promote vasculature formation like angiopoietin-1 and VEGF and induce CSCs to secrete factors involved in neoangiogenesis [46].

#### 1.2.2. Cancer-Associated Fibroblasts

Fibroblasts are one of the most important cellular components of the tumor stroma. Those that form part of the TME are called CAFs, differing from normal fibroblasts (i.e., located in non-cancerous regions) at multiple levels. CAFs, which are similar to myofibroblasts, are known to promote tumor growth, angiogenesis, and invasion [60]; however, their origin, evolution, role, and heterogeneity can differ based on the organ and/or cancer type. Regarding their origin, Mishra et al. indicated that in HCC, α-SMA^+^ stromal cells, including CAFs, could originate from MSCs as these cells showed myofibroblast characteristics in vitro, such as the expression of α-SMA. When the authors exposed human MSCs to tumor cell-conditioned medium for 30 days, markers of CAFs (such as vimentin, FSP, α-SMA, or CXCL12) increased, supporting the hypothesis of tumor cell-mediated MSC differentiation towards a protumoral CAF state [61]. Consistent with this, CAFs can increase the CSC population through prostaglandin E2 and IL-6 signaling; however, these cytokines are necessary but not sufficient to induce cancer stemness, and the activation of further signaling pathways is necessary [62]. Like MSCs, CAFs have also been shown to contribute to CSC chemoresistance. CAF-conditioned medium was shown to promote self-renewal in CRC cells, protecting them from chemotherapeutic agents like 5-fluoruracil (5-FU) or oxaliplatin (OXA). The mechanism mediating this chemoresistance was linked to exosomes secreted by CAFs, which in addition to chemoresistance, also promoted stemness by activating the Wnt signaling pathway [19,63]. Other CAF-associated pathways or factors have also been described as mediators of CSC enrichment, like STAT3-mediated CCL2 production by CAFs, which promotes self-renewal, stemness, and *NOTCH1* activation in breast cancer. Reversely, knock-down of fibroblast-derived CCL2 or its depletion with neutralizing antibodies inhibits tumorigenesis in NSG mice [64]. In addition, the IGF-II/IGF1R signaling pathway described in CAFs by Chen et al. was shown to induce *NANOG* expression and activated stemness in non-small cell lung cancer (NSCLC) cells [65]. Thus, CAFs derived from MSCs retain and acquire CSC support mechanisms while promoting TME remodeling.

CAFs have been described as predominantly proinflammatory cells across a wide variety of tumors, including PDAC, breast cancer, and squamous cell carcinoma [62,66]. Secreting a myriad of cytokines, chemoattractants, and other factors, CAFs have been shown to enhance angiogenesis, tumor growth, and macrophage recruitment to the TME [66]. Despite having many functions in common, four CAF subsets with specific spatial distribution have been identified in human samples. One of these subsets, CAF-S1, has been associated with the accumulation of FOXP3^+^ T lymphocytes and CD4^+^CD25^+^ T lymphocytes in the tumor, both populations related with cancer cell stemness as explained in the next section, while CAF-S4 has been associated with an increase of stromal cells [67]. The heterogeneity of CAFs and how they promote stemness in different cancers types (i.e., lung and breast) has been previously reviewed [68], pointing out that some subpopulations (e.g., CD10^+^GPR77^+^ secretory CAFs) enhance stem features through cytokine secretion [69], while others inhibit stemness by secreting BMP-4, as in oral carcinoma [70].

#### 1.2.3. The Immune Component

Tumors are highly infiltrated by immune cells, and the degree of infiltration and type (or polarization) of the cells can determine the tumor’s immune status or classification: dividing them into “hot” or “cold” tumors, which show high or low immune infiltration, respectively. Cold tumors are of extreme interest, and much research has been invested in determining how to convert cold tumors, which are clinically less accessible to chemo- and immunotherapy, into hot tumors that offer more therapeutic options [71]. To achieve this, however, it is important to understand that the tumor and the immune system co-evolve in a process called “immunoediting”, where a tumor changes its state from immune system vulnerable to immune system resistant, via three phases known as the three “E” rule: Elimination, Equilibrium, and Escape phases. In each phase, different immune cells are thought to participate in the immunoediting process of the CSC microenvironment, many of which are shared between most cancer types, and include cytotoxic T lymphocytes (CTLs), T regulatory cells (Treg), natural killer (NK) cells, macrophages, myeloid-derived suppressor cells (MDSCs) and T helper cells (Th), as well as some other tissue-specific cells, such as tissue-resident innate lymphoid cells [72].

T cells play many roles within and outside the CSC niche, and one can find protumoral and anti-tumoral subsets in spatially distinct zones. Activation of CD8+ T cells by antigen-presenting cells gives rise to CTLs, an anti-tumoral subset of CD3^+^CD8^+^FasL^+^ T cells. Their activation, which is based on two signals (TCR and MHC class I), may be impaired by CSCs downregulating their MHC class I molecules and upregulating those molecules that inhibit activation, such as PD-L1 or immunosuppressive cytokines. This is currently a major research focus of many laboratories seeking to improve immunotherapeutic approaches and response by establishing immune checkpoint inhibitors as a new therapeutic option [73,74]. Apart from modulating immune activation and inhibiting receptors and molecules, tumors can avoid interaction with immune cells by using the stroma as a barrier against CTL infiltration [75]. Regulatory T cells, however, can exert immunosuppressive effects through the secretion of factors like TGF-β and can thus act in a protumoral manner. Tregs are CD3^+^ CD4^+^ CD25^+^ FOXP3^+^ CD127^low^ T cells that are recruited by cancer cells via surface ligands like CCL22 and CCL35. Moreover, those which present CCR4 on their surface membrane are known as effector Tregs and are recruited by other tumor-associated cells, such as CAFs [51,76]. Tregs are recruited early during tumorigenesis and expand their population throughout the lifespan of the tumor, inhibiting the recruitment of CTLs and promoting the CSC compartment via secreted cytokines like IL6, CCL2, and TGF-β [72]. Once chronic inflammation is established in the tumor, Th cells promote the M2 polarization of macrophages by secretion of a myriad of cytokines including IL-4, IL-10, and IL-13 [77,78].

As part of the innate immune system, NK cells play an important role in tumor immunosurveillance, killing cells which express malignant traits. However, even if this process is effective in the early stages of tumor development, long-term response is disrupted by tumor immunoevasive mechanisms [79]. NK cells present in the TME are usually defined as CD3^−^CD56^+^, and they induce apoptosis of cells with reduced presentation of MHC class I molecules [80]. It has been reported that glioblastoma (GBM) CSCs are vulnerable to IL-2/IL-15-activated NK cells due to their expression of cytotoxicity receptors and ligands (i.e., PVR and Nectin-2) and the reduction of MHC-I molecules on their surface [81]. According to this, Ames et al. described the ability of NK cells to kill CSCs from breast, glioma, sarcoma, and pancreatic cancers due to CSCs expressing the NKG2D ligands MicA/B and receptors Fas and DR5. Related to this, they also reported a NK cell-mediated reduction of the CSC compartment in a metastatic breast cancer mouse model [82]. NKG2D is a crucial activating receptor expressed mostly in cytotoxic immune cells that enables the killing of stressed cells, including tumor cells [83]. However, it has been reported in leukemia that the lack of expression of NKG2D ligands in tumor cells could serve as an immunoevasive mechanism and even as a leukemia CSC marker, as NKG2D^−^ cells showed a more stem-like phenotype, including enhanced chemoresistance [84]. Another mechanism by which CSCs evade NK cells is by promoting the production of TGF-β by TME stroma cells, which would lead to reduced expression of NKG2D on NK cells, disabling the interaction between this receptor and the possible ligands expressed by CSCs [10,85].

The role of MDSCs in cancer has gained momentum in recent years due to their immunomodulatory role in the TME and their value as a possible predictive marker. There are two large populations of MDSCs, one similar to monocytes (M-MDSCs) and the other more similar to neutrophils (PMN-MDSCs). PMN-MDSCs are CD11b^+^ CD14^−^ CD15^+^ CD33^+^, while M-MDSCs are CD11b^+^ CD14^+^ CD15^−^ CD33^+^ HLA-DR^−/low^ [86,87]. Although their immunosuppressive mechanisms are similar, M-MDSCs are much more potent than PMN-MDSCs [88]. Moreover, PMN-MDSCs largely expand their population while maintaining their phenotype, whereas M-MDSCs do not undergo such a marked expansion and differentiate into TAMs [88]. Despite these differences, the presence of both MDSCs populations in tumors correlates with poor outcome in patients [89,90]. In contrast to disease-free conditions, in which myeloid-derived cells normally develop their activity, cancer-associated chronic inflammation significantly changes the cytokine landscape. This results in higher levels of protumoral signals such as TNF-α, altering the activating signals of MDSCs, and conferring upon them an immature phenotype, which is associated with a longer lifespan and reduced mobility [87]. This immature phenotype results in both types of MDSCs being present, with decreased phagocytic ability and enhanced secretion of prostaglandin E2, arginase, and anti-inflammatory cytokines [90]. MDSCs can be recruited to the TME by cancer cells via secretion of VEGF and G-CSF, which would then aid CSCs in evading detection by CTLs and NK cells [91]. Moreover, MDSCs can also attract other immunosuppressive cells like Tregs, which would also support CSC maintenance [92]. While MDSCs can secrete many factors and induce the upregulation of many transcriptional profiles in target cells, the precise manner by which MDSCs affect CSCs is still unclear. One putative mechanism is upregulation of piwi-interacting RNAs (piRNAs): the PMN-MDSC subpopulation in patients with multiple myeloma can increase the CSC compartment via upregulation of piRNA-823, a small non-coding RNA that contributes to multiple myeloma cell proliferation. This piRNA can activate DNA methyltransferase 3B both in vitro and in vivo, which has been proposed as a possible mechanism for this phenomenon [93]. Other mechanisms by which MDSCs potentiate the stem-like properties of CSCs certainly exist and may be cancer type-specific; therefore, a deeper understanding of these suppressor cells and their communication with CSCs will hopefully allow inhibition of this crosstalk in the future.

TAMs are the most abundant immune cells present within the TME of different cancer entities, and among their many functions, TAMs have been closely associated with and linked to CSC support and maintenance. The role of TAMs in the CSC niche has been extensively reviewed [85,94,95,96,97,98]. Macrophages are usually characterized by the expression of CD14, CD68, CD115, CD312, or HLA-DR [99]. However, different subsets of macrophages with distinct phenotypic and genotypic profiles have been described and, for a lack of consensus in the field, macrophages are generally, and perhaps too easily, divided into two polarization states: M1 and M2. The first subset is proinflammatory and is involved in antitumoral responses, in part by attracting other components of the immune system, like CTLs or NK cells, to the tumor. On the contrary, M2 macrophages are characterized by the expression of an anti-inflammatory signature of cytokines and the expression of distinct cell-surface markers like CD163, CD169, CD204, CD206, or Tim-3 [95,100]. While M2 macrophages have been associated to and extensively described in wound healing, M2-like macrophages can also suppress anti-tumor immune cell populations in cancer [97], further potentiating the immunosuppressive tumor state. Thus, it is not surprising that TAMs closely resemble M2 macrophages, and often these labels are inaccurately interchanged to define the macrophage populations present within the TME [95]. However, consensus exists with regard to the capacity of TAMs to communicate and activate the CSC compartment [95]. Our group recently identified two essential pathways by which TAMs can activate PaCSCs: First, we observed that IFN-stimulated gene 15 (ISG15) is secreted by TAMs in response to IFN-β produced by CSCs, thus establishing a paracrine signaling loop that reinforces the CSC phenotype in vitro and in vivo [101]. In addition to secreted ISG15, we identified cationic antimicrobial protein 18 (hCAP18/LL-37) as another protein involved in the TAM-CSC crosstalk. LL-37 is produced by TAMs in response to CSC-secreted TGF-β and Nodal/Activin. It is recognized by both formyl peptide receptor 2 (FPR2) and P2X purinoceptor 7 receptor (P2X7R) present on the surface of CSCs, activating intracellular signaling pathways that enhance key CSC phenotypes, such as tumorigenesis. Importantly, when FPR2 or P2X7R were inhibited, a strong reduction in tumor size and circulating tumor cells was observed in several in vivo models of PDAC [102]. TAM-secreted factors can also enhance other CSC-related phenotypes, such as chemoresistance. Proteomic analysis of macrophages co-cultured with apoptotic PDAC cells identified 14-3-3ζ, a member of a larger family of anti-apoptotic proteins, which we showed could promote PDAC cell survival through an Axl-mediated signaling pathway. Inhibition of this protein in combination with gemcitabine treatment resulted in increased chemosensitivity, suggesting that blocking 14-3-3ζ may be therapeutically beneficial [103]. In NSCLC CSCs, it has been described that TAMs, like other CSC niche components, secrete a variety of interleukins, including IL-10, which can promote CSC properties through Jak1/STAT1/NF-κB/Notch1 signaling [104]. In lung cancer, MUC1 has been found to be overexpressed in patients and correlates with poor prognosis. Inhibition of MUC1 with pterostilbene abolished the polarization of macrophages towards an M2 phenotype and reduced the CD133^+^ CSC population via the downregulation of NF-κB and p65 signaling [105]. GBM-associated microglia, a specific type of tissue-resident macrophage in the brain, has been shown to promote the expression of stemness markers in GBM cells, including *NANOG*, *SOX2*, *ALDH1*, *OCT3/4*, and *BMI1*. Microglia mostly originate from bone marrow-derived myeloid cells and, in GBM, they can establish a protumoral immune population in the so-called “border niche” [106].

#### 1.2.4. Vasculature

In order to develop and grow, tumors require nutrients delivered by blood and lymphatic vessels. Vasculature is an important part of the TME, and it is mainly created by nascent endothelial tissue ramifications that grow into the tumor bulk and that are stimulated by factors secreted by CSCs and other cancer cells. These factors, most importantly VEGF, also recruit endothelial progenitor cells (EPCs), which migrate from the blood to the TME to build the aberrant neovasculature [107]. The hypoxic conditions that are found within the tumor mass also play a key role in the promotion of the vascular network. HIF-1α, a major hypoxia response factor and regulator of stemness in CSCs, is expressed in low oxygen conditions, and its expression correlates directly with VEGF production [108,109]. MicroRNA-21 has been described as another factor that could induce VEGF expression and its secretion through exosomes in glioma CSCs (gCSCs) [110]. Importantly, lymphatic vessels are different from blood vessels in terms of structure (the former do not show pericytes nor smooth muscle cells supporting the structure) and growth-inducing signals, although angiopoietins and VEGF also play a role in their development [111].

The crosstalk between CSCs and the neovasculature constitutes a bidirectional highway where CSCs induce the creation of new vessels, and the vasculature cells further promote the stem-like properties of cancer cells and enhance their radio- and chemoresistance [112,113,114]. Vascular endothelial cells (ECs) have a direct role in promoting cancer cell stemness, as tumor vasculature ablation leads to a reduction in the CSC population. Interestingly, the induction of vessel formation promotes tumor growth in vivo and co-culturing of tumor cells with ECs in vitro increases the self-renewal capacity of tumor cells [115,116]. Moreover, dedifferentiation of glioma cancer cells into stem-like cells was observed in the presence of EC conditioned medium. This phenomenon was traced back to basic fibroblast growth factor (bFGF) secreted by ECs in culture [117]. In less aggressive tumors, the same conversion (i.e., acquisition of a stem-like state) has been described both in vitro and in vivo through the upregulation of IGF1 in ECs by tumor cells, which in turn activates the FGF4-FGFR1-ETS2 pathway, resulting in a more aggressive phenotype linked to increased metastatic capacity [118]. Moreover, differentiation of CSCs into other cell types when they are in contact with ECs has been reported. Subpopulations of CSCs are located in perivascular niches within the tumor, and they possibly contribute to stabilizing aberrant blood vessel structures, in part by differentiating into pericyte-like cells, which further underscores the wide plastic range of CSCs [119]. It has also been proposed that CSCs are able to acquire an endothelial-like phenotype forming new vasculature themselves. This vasculogenic mimicry observed across different types of tumors (e.g., breast cancer, uveal melanoma, and neuroblastoma) results in the generation of an open lumen, allowing blood cells to circulate. This angiogenesis-like mechanism shows independency from VEGF and instead relies on CSC plasticity, creating a possible anti-VEGF therapy resistance system that would benefit tumor progression and the CSC compartment [107,113,120,121].

## 2. CXCL12 Axes and Their Role in Cancer

Chemokines are a superfamily of cytokines that participate in numerous biological processes at the cellular level. They act as chemoattractants, binding specific G-coupled-protein 7-spanning transmembrane receptors and, based on the distribution of the first two conserved cysteines in their structure, they can be classified into different subfamilies, each of which possesses one of the following motifs: CXC, CC, XC, and CX_3_C. Chemokines can promote a myriad of physiological processes including homeostasis, cellular trafficking, cell adhesion, vessel growth, and stem cell support [122,123,124,125]. CXCL12, also known as stromal derived factor 1 (SDF-1), is a chemokine needed by the hematopoietic stem cell (HSC) niche to ensure HSC survival, proliferation, and for the homing of leukocytes. CXCL12 was first described in 1993 in a cDNA cloning study [126] and was cloned and structurally defined afterwards [127]. In 1996, Nagasawa et al. identified CXCL12 as an important mediator of the proliferation of B cell progenitors in vitro and showed its expression in bone marrow progenitor cells. The same study also reported that knockout of CXCL12 in mice was lethal, with mice dying perinatally due the developmental failure of the immune and cardiac systems [128]. Numerous studies have since described CXCL12 in a wide variety of processes and diseases such as HIV [129,130], rheumatoid arthritis, asthma, amyotrophic lateral sclerosis [131], and cancer [132]. In cancer, CXCL12 has been studied in depth in various tumors (Table 1), and is expressed in cancer cells and most of the cells of the TME, underscoring its role as a main player in metastasis [124].

Although the focus has been mainly on CXCL12, its receptors have also attracted wide attention of researchers. CXCR4 is the most commonly expressed receptor in human cancers and is a surface marker that serves to identify CSCs of many tumor types [141,147]. Moreover, its relevance is rooted in the fact that CXCL12 was initially believed to be its only ligand [148,149]; however, other receptors (e.g., CXCR7) that bind CXCL12 have been discovered [150]. Thus, CXCL12 has multiple functions in tumor biology including (a) promotion of cell survival, cell invasion, and CSCs features; (b) recruitment of stromal cells to the TME to promote tumor growth and immune evasion; and (c) promotion of angiogenesis [151]. A diagram comparing the effects caused by CXCL12 through the interaction with CXCR4 or CXCR7 on CSCs is detailed in Figure 2.

### 2.1. The CXCL12/CXCR4 Axis

The CXCL12/CXCR4 axis constitutes one of the most important signaling pathways that regulates tumor growth, angiogenesis, and metastasis, and it is a predictive factor for survival in patients [152]. CXCR4 is primarily located on the cell membrane; however, the interaction with CXCL12 induces the internalization of CXCR4, which is the reason why CXCR4 has been observed in different subcellular compartments [153]. These different subcellular localizations have been observed in breast cancer primary tumor cells (where it was membrane-bound) and in lymph node metastasis (where it was cytoplasmic) [154]. The CXCL12/CXCR4 axis is regulated in multiple ways. CXCL12 can regulate its own expression via an autocrine loop of epigenetic modifications. DNA methyltransferases 1 and 3B can hypermethylate the promoter region of *CXCL12*, leading to a downregulation of *CXCL12* mRNA expression [155]. Methylation patterns also play an important role in *CXCR4* expression, the lack of them being associated positively with tumor size, stage, lymph node status, metastasis, and *CXCL12* promoter region methylation. The same study showed that cell lines expressing CXCR4 showed 15% methylation of CpG dinucleotides in the *CXCR4* promoter region, while the lack of *CXCR4* expression was due to a high density of methylation (around 91%). Altogether, these data were used to conclude that the methylation status of *CXCR4* or *CXCL12* could be used as a potential biomarker for patient prognosis [156].

Transcription factors like c-Myb, Slug, and Akt1 and molecules like HIF2, p53, and FOXP3 have been shown to regulate the CXCL12/CXCR4 axis in different ways. Slug, Akt1, c-Myb, and HIF2 can increase the expression of *CXCL12*, while p53 and FOXP3 may reduce its expression [157,158,159]. MicroRNAs also play a role in the post-transcriptional modulation of *CXCL12* expression [153], and *CXCR4* can be induced at the transcriptional level by HIF-1α together with NF-κB, which suggests that this axis is upregulated under hypoxic conditions [160]. At the CSC level, some markers like CD24 have been shown to upregulate *CXCR4* [161], and in PDAC, CXCR4^+^CD133^+^ CSCs were first demonstrated to identify a specific CSC subset with exclusive metastatic potential [147]. CXCR4^+^ CSC populations have also been described in other tumors like breast [162], gastric [163], renal [164], and colorectal [3] cancers. Inhibition of the CXCL12/CXCR4 axis has been reported to result in a reduction in STAT3 and ERK phosphorylation, decreasing the PaCSC population [165]. How this axis is expressed in different CSC populations as well as in stromal cells of other tumors, the parallels between the processes regulated by this axis, and its roles in the homeostatic HSC and CSC niche will be discussed later.

### 2.2. The CXCL12/CXCR7 Axis

Until very recently, CXCR4 was thought to be the only receptor to bind CXCL12; however, CXCR7 has emerged as another receptor for CXCL12, with implications in cancer development. CXCR7 has been found to be membrane-bound in many tissues and cell types, including cancer cells, ECs, and immune cells, and it is implicated in cellular proliferation and adhesion, contributing to biological processes, including, but not limited to, tumor growth and embryonic development [166,167]. Alterations in the CXCL12/CXCR7 axis can influence tumor cell invasion and adhesion processes in prostate cancer, increasing the expression of CD44, and niche factors like VEGF, which suggests a role for CXCR7 in the CSC niche [168]. Accordingly, Tang et al. have also explored the connection between CXCR7 and the stem-like phenotype of breast cancer cells, showing that shRNA-mediated inhibition of CXCR7 led to a reduction in CD44 expression and in other stem cell markers such as NANOG or ALDH1. Moreover, combined treatment of xenograft-bearing mice with shRNAs targeting CXCR7 and epirubicin displayed an enhanced antitumoral effect compared to single treatments [169]. CXCR7 expression has also been evaluated as a prognostic factor in cancer, and a positive correlation between CXCR7 and tumor size, grade and differentiation has been described in PDAC [170,171]. Similar results have been generated for prostate [172] and CRC [173,174]. However, in diffuse large B cell lymphoma, co-expression of CXCR4 and CXCR7 is associated with a higher survival rate, while CXCR7+/CXCR4- patients show a poorer survival [175]. These results suggest a possible interaction between the CXCL12/CXCR7 and CXCL12/CXCR4 axes with consequences at the level of cancer aggressiveness, something that has also been studied in vitro in CRC and prostate cancer cells overexpressing both receptors [168]. Specifically, a study by Heckmann et al. showed that the gene expression regulation of both receptors is partly opposite, with *CXCR4* expression being dominant [176]. MicroRNAs, hormones and microenvironmental factors also play an important role in the regulation of *CXCR7* expression. In prostate cancer, *CXCR7* expression is sensitive to androgen levels, as high hormone levels induced a reduction in the levels of the receptor [177]. MicroRNA-100 is suspected of binding the 3′-untranslated region of the CXCR7 mRNA, impeding translation initiation and thus decreasing the overall levels of CXCR7. Indeed, in gastric cancer, the levels of both are inversely correlated [178]. Another example is IL-6, which is secreted by CAFs and other TME cells and constitutes a key player in the CSC niche. IL-6 upregulates CXCR7 expression via STAT3/NF-κB signaling, and this upregulation may serve as a mechanism for cisplatin resistance development and for the increased expression of stemness markers in esophageal cancer cells [179].

### 2.3. The CSC Niche Mimics the HSC Niche

Possibly, one of the most studied homeostatic niches is the HSC niche in the bone marrow. While the biology of HSCs has been widely reviewed [180,181], the dynamics of the HSC niche remain elusive. The same can be said for the CSC niche, which mimics many of the biological and structural features of the HSC niche. As early as 2006, Scadden recognized the importance of the HSC niche and postulated that “if cancer is organized in a manner similar to normal tissue, with a minor subpopulation of stem cells, an attendant blood supply and a unique microenvironment, there might be a similar dependence of the stem cells on a cancer niche” [182]. Thus, to understand the CSC niche, many researchers have sought a better understanding of the HSC niche. In this context, the identification of CXCL12 abundant reticular cells (CARCs) in the HSC niche revealed that CXCL12 ensures HSC maintenance and retention in the bone marrow [183]. CXCL12 had been previously identified as an important chemokine regulating HSCs and bone marrow homeostasis, but the discovery of CARCs helped to understand the importance of the CXCL12/CXCR4 axis in HSC differentiation and how the microenvironment responds to external stimuli [184]. As detailed above, MSCs play an important role in the regulation of the CSC niche. MSCs are recruited from the bone marrow, where they play a similar role in HSC niche support [185]. In hematological cancers, myelofibrosis alters the bone marrow stroma, increasing the activity of MSCs, which differentiate into stromal cell-like osteoblasts, favoring the progression of leukemic stem cells [186]. The influence of hypoxia is also a commonality shared between CSCs and HSCs, both of them with populations residing in hypoxic niches and expressing HIF-1α [119,187,188]. Moreover, a considerable part of the cellular components of the CSC niche seems to come from the bone marrow. HSCs migrate through different hematopoietic niches during ageing, but stimuli such as tissue damage can induce this “homing” and the mobilization of other bone marrow-derived cells (BMDCs). In cancer, CXCL12 regulates homing via the establishment of a hypoxia-dependent gradient that drives BMDCs to the CSC niche in the primary tumor or by building up the metastatic niche where needed, to ensure CSC growth and the generation of new vasculature [189]. Studies in GBM have shown that CSCs and HSCs use the same mechanisms to ensure their retention in the niche. Not only the interaction of CXCR4 with CXCL12, but also the interaction of osteopontin with CD44 regulate the localization of CSCs in the niche [29,190]. In AML, the migration of leukemic stem cells is also guided through the CXCL12/CXCR4/CXCR7 axes [191]. These similarities should be considered in the process of identifying new targets and in the development of new therapies to target CSCs and disrupt the CSC niche.

## 3. CXCL12 in Different CSCs

### 3.1. Breast Cancer

A CXCL12/CXCR4 interaction drives breast cancer metastasis, as evidenced by a direct correlation between high CXCR4 levels in primary tumors and high metastatic capacity [192]. In primary tumors, CXCR4 was upregulated, while CXCL12 was downregulated due to hypermethylation of the *CXCL12* promoter region. This downregulation was associated with lymph node metastasis and lack of estrogen receptor (ER) expression [155]. Interestingly, CXCL12 expression is linked to ER expression in ER-positive breast cancer, regulated in this case by MSCs. Inhibition of both CXCL12 and ER resulted in decreased proliferation and migration [193], suggesting that CXCL12 differentially regulates proliferation and migration in breast cancer depending on the expression levels of ER. Importantly, CXCL12 concentrations in the TME seem to be a key factor, as different CXCL12 levels activate different Rho GTPases. When CXCL12 expression is low, breast cancer cells promote migration via activating RhoA, while high concentrations allow the interaction between RhoA and Rac1, promoting cell–cell and cell–matrix adhesion and thus inhibiting migration [133]. CXCL12 overexpression induces EMT in breast cancer cells, activating stem cell-associated genes like *SOX2*, *OCT4*, and *NANOG*, promoting a stem-like phenotype marked by ALDH1 expression and Wnt/β-catenin pathway activation [194]. The activation of NF-κB signaling via CXCL12 is another means of activating EMT, resulting in an enrichment of the CSC population [135]. Sphere formation, a routinely used technique to measure stemness and self-renewal, was used to show that CXCR4 overexpressing breast cancer cells had a higher self-renewal capacity when cultured in hypoxia. Furthermore, the side population (SP), a group of cells with stem-like features that are able to extrude chemotherapeutic agents and live cell dyes through drug-efflux pumps such as ABCG2, increased and showed an overactivation of the oncogenic JNK/cJun pathway, suggesting that cJun could mediate the transcriptional activation of *ABCG2* by binding its promoter, thus increasing the sphere formation capacity of SP cells overexpressing CXCR4 [134].

### 3.2. Lung Cancer

Similar to healthy breast tissue, healthy lung cells express CXCL12 at low to undetectable levels as compared to tumor tissue [195,196]. In NSCLC, CXCR4^+^ chemotherapy-resistant cells display increased self-renewal capacity and CXCL12 secretion. STAT3, mTOR, and Akt signaling pathways seemed to play an important role in this induction; however, STAT3 inhibition did not affect CXCR4 expression, indicating that STAT3 is a downstream component of CXCL12/CXCR4 signaling [136]. Another study in Lewis lung cancer cells identified a subset of CXCR4^+^ cells that were proangiogenic and pro-metastatic, and showed overexpression of other stemness-related proteins such as VEGF or MMP9 [197]. In vivo studies in mouse models and patient-derived xenografts (PDX) also revealed CXCR4^+^ stem-like cells in NSCLC. Interestingly, CXCR4 expression was restricted to CD133^+^ cells and conferred chemoresistance [4]. This finding was supported by another study in lung PDX models, in which Bertolini et al. additionally discovered that lung CSCs with metastatic ability were EpCAM^−^. The authors observed that the TME had a significant influence on EMT induction in CSCs, resulting in increased expression of CD133 and CXCR4, and decreased expression of EpCAM. CXCR4 led metastatic progression of lung CSCs, as selective inhibition of CXCR4 with CTCE-9800 reduced the migratory ability of CXCR4^+^ CSCs. The role of CXCL12 secreted by stromal cells like CAFs was also addressed and confirmed, highlighting its influence on the induction of EMT in lung CSCs through its interaction with CXCR4 [198,199]. CXCR4 has also been shown to play a role in lung CSC chemoresistance. Xie et al. showed that CXCR4^+^ lung CSCs were more resistant to cisplatin due to the upregulation of the cytochrome p450-associated molecule CYP1B1, whose expression positively correlated with CXCR4 [137]. The authors later showed that these CXCR4^+^ cells expressed CD133, and that CD133 mediated the expression of CXCR4 [199]. As mentioned before, hypoxia regulates CSC fate, and promotes EMT and the acquisition of a stem-like state in lung cancer cells [200]. It is well established that CXCR4 is overexpressed in hypoxia, not only through the activation of signaling pathways like NF-κB [201], but also via epigenetic mechanisms, like demethylation of the *CXCR4* promoter [202]. TGFβ, a widely studied molecule with an important role in regulating the TME, increases the expression of CXCR7, CXCR4, and CXCL12 in lung adenocarcinoma cells. Wu et al. were able to demonstrate that CXCL12 contributes to the maintenance of lung adenocarcinoma cell self-renewal, and that CXCR7 and CD44 levels positively correlate with CXCL12 expression. This study highlighted the important influence of TME-derived factors in the regulation of the CXCL12/CXCR7 axis, and how this can influence the acquisition of CSC-like phenotypes in lung cancer [203].

### 3.3. Squamous Cell Carcinoma

The role of the CXCL12/CXCR4 axis in squamous cell carcinomas (SCC) has not been characterized to date with the same level of detail as in other types of cancer. However, studies using cell lines and patient samples have shed light on its role in some SCC-associated characteristics. Although head and neck squamous cell carcinoma (HNSCC) has a good outcome with surgical intervention, a stem-like chemoresistant cell population that is able to drive relapse has also been identified in HNSCC using CSC cell surface markers such as CD44 [204]. Using this marker, Faber et al. showed that CXCL12 did not have an impact on proliferation in the CD44^+^ subset that was also CXCR4^+^, but increased podia formation, thus promoting the migratory capacity of cells [138]. Interestingly, an analysis of patient samples showed that CXCR4^+^ cancer cells were located at the invasive front of the tumors, in intimate contact with the stroma [205].

In mouse models, advanced skin SCCs undergo an expansion of the CSC subset due to the autocrine activation of Pdgfrα. These CSCs express Cxcr4, Cxcr7, and secrete Cxcl12. In contrast to what had been previously described by Faber et al., this activated cell proliferation and survival and promoted lung metastasis [139]. The authors validated their findings in patient samples and reported the upregulation of PDGFRα, PDGFRβ, and CXCL12 in tumor cells. Importantly, inhibition of the interaction between these cytokines and their receptors blocked metastasis. Thus, the CXCL12/CXCR4/CXCR7 axes likely do play an important role in SSC, although further research is still needed.

### 3.4. Gastrointestinal Tumors

In gastric cancer, CXCR4^+^ CSCs are associated with tumor formation and ascites accumulation, serving as a tool for patient stratification [163]. On the other hand, Hayakawa et al. have described a CXCR4^−^ Mist1^+^ gastric CSC population in which the CXC12/CXCR4 axis regulates the TME, where endothelial cells produce CXCL12 and the innate lymphoid cells (ILCs) express CXCR4. This perivascular axis induces Wnt5a secretion by ILCs and promotes RhoA activation in tumor cells, which promotes sphere formation [140]. Esophageal CSCs secrete CXCL12 and express CXCR4, which promotes the formation of an autocrine signaling loop that activates the ERK1/2 pathway and serves as a system to maintain the invasive, metastatic, and stem-like properties of these CSCs [141]. This axis can also be integrated into a more complex signaling pathway, as it occurs in HCC. Among all the risk factors associated with HCC development, hepatitis B virus (HBV) infection is also linked to the CXCL12/CXCR4 axis. It has been reported that the HBV protein X (HBx) has a role in maintaining the hepatic OV-6 antibody (OV-6)^+^ CSC population by binding the mouse double minute 2 homolog (MDM2) and inhibiting its degradation. In this scenario, MDM2 induced CXCL12 and CXCR4 expression in a p53-independent manner and also activated the Wnt/β-catenin pathway, enhancing OV-6+ stem-like properties [142]. CSCs are well known to be highly chemoresistant and, according to the CSC model, they survive therapeutic assaults and facilitate tumor cell repopulation after treatment. In HCC, CD133^+^/CD24^−^ cells, which are resistant to ionizing radiation, undergo a series of epigenetic modifications that leads to the overexpression of CXCL12. Methylation of H3K4 and demethylation of the *CXCL12* promoter region in H3K9, together with the recruitment of the c/EBPβ transcriptional activator, promoted the activation of the CXCL12/CXCR4/CXCR7 axes and the acquisition of a more invasive and aggressive phenotype in radioresistant cells in response to radiation [206]. In PDAC, we identified CXCR4^+^CD133^+^ as an exclusively metastatic subpopulation of PaCSCs from a highly metastatic cell line (L3.6pl), derived from the COLO 375 cell line, isolated after successive cycles of selection in nude mice [147]. However, the role of CXCL12 in PDAC remains controversial, beginning with its expression levels. Some studies support the idea that CXCL12 is silenced or expressed to low levels in PDAC cells. One study showed that administration of CXCL12 to PDAC cells, in which CXCL12 expression had been silenced, reduced their oxidative phosphorylation, glycolytic capacity, and dissemination in a mouse model [207]. In addition, microRNA-454 has been identified as a key regulator of CXCL12 in PDAC via biding the 3′-UTR of CXCL12 mRNA. To test the consequence of this interaction in vivo, NOD/SCID mice were injected with established human PDAC cell lines with reduced CXCL12 expression concomitant with microRNA-454 knock-down or overexpression, respectively. While silencing of microRNA-454 led to increased CXCL12 expression, macrophage recruitment via the CXCL12/CXCR4 axis, and a reduction in tumor size, the overexpression of microRNA-454 significantly reduced CXCL12 expression and increased tumor size, indicating that CXCL12 negatively impacts PDAC progression [208]. On the other hand, KLF10, a factor that is involved in the regulation of TGFβ, is lost in more aggressive and metastatic PDAC and seems to have an impact on CXCL12 expression in PDAC cells. KLF10 loss is associated with overexpression of CXCL12 and CXCR4, which promotes metastasis and a CSC phenotype by activating c-Jun, supporting the idea that CXCL12 is indeed an important player in PDAC, although its role is still not clear [143]. The same phenomenon has been reported in CRC, in which CXCL12/CXCR4 is a key axis that induces cell migration and metastasis of CSCs [209]. In contrast, silencing of the *CXCL12* promoter by hypermethylation has also been reported to promote metastasis in CRC, while reactivation of CXCL12 expression reduced metastatic spread [210]. Along this line, another study described CD44v6 as a CSC marker in CRC and showed that its expression seemed to be induced by factors like osteopontin and hepatocyte growth factor (HGF), which promoted CXCL12/CXCR4 axis activation and induced a more metastatic phenotype in the CRC CSC population [24]. A comprehensive review of the role of the CXCL12/CXCR4/CXCR7 axes in gastrointestinal tumors was recently published by Daniel et al. [211].

### 3.5. Brain Tumors

One of the most studied CSC populations are the CSCs of brain tumors, such as glioblastoma multiforme (GBM). Glioma cancer stem cells (gCSCs) reside in a niche close to CD31^+^ ECs and express CSC markers like CD133 and more brain-specific markers like Nestin. The gCSC niche shares major features with the HSC niche, including an important role of CXCR4 and CXCL12 in the niche [29]. For both GBM and glioma, CXCL12 functions under hypoxic conditions in the CSC niche through an autocrine positive feedback loop that promotes cell survival, tumor progression, and cell proliferation, which can be inhibited by blocking CXCR4 [144,145]. Importantly, interstitial fluid flow is a driver of GBM stem cell (GSC) migration due to the chemotaxis induced by CXCL12/CXCR4 and CD44. While radiotherapy is part of clinical treatment in brain tumors, radiation has been reported to induce invasion, which has been linked to an increase of radioresistant CXCR4^+^CXCL12^+^ GSCs [212]. As expected, CXCR7 expression has been described in both gCSCs and GSCs and reported to have the same function as CXCR4 [213,214]. Other CSC niche factors have substantial influence on the regulation of CXCL12 function, such as cathepsin K, a protease that cleaves CXCL12 to form inactive fragments. Thus, inhibition of cathepsin K would promote the chemotactic activity of CXCL12 [190,214]. Moreover, *NOTCH1*, characteristically overexpressed in gCSCs, has been reported to be a modulator of CXCL12/CXCR4 expression via the PI3K/AKT/mTOR signaling pathway. Interestingly, shRNA-mediated *NOTCH1* knock-down resulted in reduced tumor growth, self-renewal, and invasiveness in a model of GBM [20]. Studies focused on the destination(s) of migratory GSCs have identified the subventricular zone as a permissive niche that can be colonized. In a study by Goffart et al., gCSC migration followed the CXCL12/CXCR4 axis [215] and, once located in the subventricular zone, CXCL12^+^ gCSCs displayed a radioresistant phenotype mediated by the secretion of CXCL12 [216]. Finally, it is not surprising that one of the main processes mediated by CXCL12 in the gCSC niche is angiogenesis. gCSCs have been reported to attract bone marrow endothelial progenitors to their niche via CXCL12 secretion [217], which in turn is activated by hypoxia through the PI3/Akt pathway [218].

### 3.6. Hematological Cancers

Due to its role in homeostasis of the HSC niche, CXCL12 is likely to play a key role in hematological tumors and their respective CSCs. In AML, CSCs express CXCL12 and CXCR4, with CXCL12 being a promoter of CSC survival and tumor repopulation by regulating the interaction with the stromal microenvironment [219]. Tetraspanin 3 is a key regulator of the CXCL12/CXCR4 axis, and its suppression impaired CXCL12-dependent migration, with CXCR4 overexpression being able to reactivate this process [220]. Furthermore, osteoblasts and MSC-secreted CXCL12 were shown to interact with CXCR4 receptors of leukemia stem cells, activating survival and antiapoptotic signaling pathways, specifically MAPK and PIK3/Akt [146]. In accordance, survival of B cell chronic lymphocytic leukemia cells is dependent on CXCL12 signaling [221]. In lymphoplasmocytic lymphomas, CD20^−^ CD138^−^ CSCs expressed CXCR7 and increased in numbers in a time- and dose-dependent manner when stimulated with CXCL12. An autocrine CXCL12 signaling loop in CSCs was discovered when in hypoxia. However, a direct link between CD20 and CXCR4 has not yet been detected in patient samples [222].

## 4. CXCL12 in the CSC Niche

### 4.1. Immune System

The CSC niche components reviewed above play a crucial role in the regulation of the CXCL12/CXCR4 axis, interacting with each other and with the CSC population to sustain the tumor. One of the main roles of CXCL12 in the CSC niche is the regulation of immune cell migration in a concentration-dependent manner. When CXCL12 concentrations are low, lymphocytes, and MDSCs are attracted to the CSC niche, while high CXCL12 levels function as a chemorepellent [72,211]. Reduced CXCL12 levels in an orthotopic tumor model using the murine ovarian cancer cell line ID8-T changed the composition of the immune cell population within tumors, reducing myeloid cells, plasmacytoid dendritic cells and Tregs, while increasing IFN-γ^+^/IL10^+^ tumor-infiltrating T lymphocytes [223]. A similar result was described in PDAC mouse models with the use of trap proteins to target Cxcl12 and Pdl1. While the Cxcl12 trap induced IFN-γ^+^ T cell infiltration, the Pdl1 trap enabled these T cells to kill the tumor cells. The combination of both trap proteins changed the immunosuppressive landscape of the tumor, reducing MDSCs and M2 macrophages [224]. Thus, cancer therapies often change the immune landscape of the tumor, which may be a possible factor that influences treatment success or failure. The use of anti-VEGF antibodies in CRC upregulated CXCL12/CXCR4 signaling between cancer cells and Ly6^low^ monocytes, increasing the latter population and thus generating an immunosuppressive environment and increasing the possibility of therapeutic failure [225]. Moreover, accumulation and migration of MDSCs into the TME of osteosarcomas is mediated by a CXCL12 gradient, resulting in reduced infiltration of CTLs, leading to immune checkpoint inhibitor failure [226]. Along this line, recruitment of MDSCs via CXCL12 is an aggravating factor in breast cancer, influenced by the levels of ER [227]. The migration of macrophages, which express CXCR4, is also strongly influenced by CXCL12 concentrations [228]. Mononuclear phagocytes (i.e., monocytes, macrophages/histiocytes, and dendritic cells) are also heavily influenced by the CXCR4/CXCL12 axis. Rigo et al., showed that CXCL12-activated mononuclear phagocytes to send antiapoptotic and proliferative signals to cancer cells, while, in turn, tumor cells produced factors that polarized mononuclear phagocytes towards an M2 phenotype. The same study also showed that TAMs in CRC patient samples displayed an M2 phenotype and expressed CXCR4, with some also expressing CXCL12 [229]. The pro-tumor environment established by TAMs has also been described in other tumor types such as astrocytoma, in which CXCL12 is depicted as a critical mediator of tumor progression and TAM infiltration [230]. Moreover, the recruitment of TAMs seems to be enhanced when cancer cells interact with stroma cells, as described in multiple myeloma models. Multiple myeloma cells show increased CXCL12 production upon interaction with bone marrow stromal cells, thus promoting the recruitment and polarization of M2 TAMs, and subsequently changing the immune status of the tumor towards a more immunoevasive phenotype. Beider et al. demonstrated that CXCL12 production and CXCR4 expression in multiple myeloma cells were higher compared to healthy counterparts, and the presence of M2 TAMs was linked to a reduction of M1 TAMs and T cell infiltration, as well as IFN-γ production [228], all of which contribute to an immunosuppressive protumor environment. Furthermore, the CXCL12/CXCR4 axis is involved in cancer cell intravasation. Arwert et al. described how tumor cell-secreted TGFβ stimulates the expression of CXCR4 in TAMs. This overexpression guides TAMs along the CXCL12 gradient produced by perivascular CAFs, resulting in the acquisition of a perivascular TAM phenotype that induced vascular leaks and facilitated cancer cell migration [231].

### 4.2. Mesenchymal Stem Cells and Cancer-Associated Fibroblasts

Bone marrow and adipose tissue MSCs are recruited from their respective niches to sites of wound healing and inflammation via the CXCL12/CXCR4 axis [48]. Variations in CXCL12 expression levels are regulated in part by MSCs, and, accordingly, when CXCL12, CXCR4, or ER were blocked in a human breast carcinoma cell model, MSC-induced proliferation and migration were significantly decreased [193]. A similar phenomenon has been described in GBM in vitro and in vivo, where CD133^+^ CSCs produce CXCL12 to recruit CXCR4^+^ MSCs. This recruitment increased tumor cell proliferation, invasion and vascular growth [232]. In hematological cancers, HSCs share their niche with CSCs, so MSCs actually support both populations. Interestingly, among the CXCL12^+^ MSCs supporting CSC maintenance, Prx1^+^ MSCs negatively regulate CSC proliferation, and when CXCL12^+^ Prx1^+^ MSCs were depleted, chronic myeloid leukemia CSC self-renewal was enhanced and blood cancer cells increased proliferation and expansion, while HSC proliferation decreased. Although CXCL12 is normally associated with migration, in leukemia it has been shown to induce a quiescent state in CSCs, as well as resistance against tyrosine kinase inhibitors [233]. Another relevant contribution of MSCs in the CXCL12-dependent regulation of CSCs and their niche is through their ability to differentiate into CAFs. CAFs have been reported to recruit ECs [234] and CD4^+^CD25^+^ T cells [67] to the CSC niche in human breast cancer along the CXCL12/CXCR4 axis. As described before, colorectal CSC formation, survival, and expansion is closely linked to CXCL12 expression [24]. The same study also pointed to CAFs as the main source of the chemokines involved in the conversion of CD44v6^−^ into CD44v6^+^ migratory CSCs during tumorigenesis. Thus, MSCs play an important role in CSC maintenance through the CXCL12/CXCR4 axis regulating CSC self-renewal and expansion, as well as recruiting and/or differentiating into other TME components such as CAFs.

### 4.3. Vascular Endothelial Cells

ECs from blood and lymphatic vessels also play important roles in the CSC niche, via regulating the circulation and attraction of cells to the TME and mediating hypoxia by providing enough blood supply. In the case of hematological malignancies, contrary to what happens with MSCs, depletion of CXCL12 in BM ECs has been reported to reduce CSCs, as shown in chronic myeloid leukemia [233]. The involvement of CXCL12 in vascular metastasis of CSCs has also been described in melanoma. For example, Kim et al. showed that CXCR4^+^/CD133^+^ melanoma CSCs were guided to the lymphatic metastatic niche by CXCL12 secreted from lymphatic ECs [235]. Moreover, radiotherapy induces EMT in ECs, altering the vascular structure during relapse after treatment. This results in increased expression of CXCR4 in ECs, increased CD44v6^+^ CSC numbers, recruitment of TAMs via the CXCL12/CXCR4 axis and M2 polarization of macrophages [236].

## 5. Therapeutic Approaches to Target the CXCL12/CXCR4/CXCR7 Axes

Altogether, a plethora of data suggests that the CXCL12/CXCR4/CXCR7 axes may be potential targets for novel anti-tumoral therapies and, consequently, could directly affect CSCs and/or disrupt their niches. One of the most frequently employed drugs is the small molecule CXCR4 antagonist AMD3100 (Plerixafor). In medulloblastoma and GBM cell lines, AMD3100 blocked cell migration, survival, and proliferation, while treatment of PDXs with this drug inhibited tumor growth [237]. While AMD3100 is highly effective at blocking CXCR4, it has also shown affinity for CXCR7, and can be cardiotoxic when used as a long-term treatment in HIV patients; however, AMD3100 has been approved by the FDA and EMA as a treatment for other pathologies and could also be approved as a cancer therapy in the future [238]. Nevertheless, newer and safer drug candidates have been developed. For example, PRX177561, another CXCR4 antagonist, has been tested in GBM both in vivo and in vitro, and studies have shown that it reduces tumor growth, blocks the polarization switch of macrophages from M1 to M2 and reduces GSCs stem-like features [239]. Importantly, as a consequence of the discovery of the role of CXCR4 in HIV infection, synthetic polypeptides have been developed. TN1, a peptide that simulates the CXCL12 recognition site for CXCR4 and acts as a competitive antagonist, was modified to reduce its toxicity and increase its stability and is now used for breast cancer treatment. TN14003, the modified form of TN1, reduced cell invasion in vitro and blocked metastasis in breast cancer mouse models, generating promising results [192]. Interestingly, an oncolytic virus-based approach, consisting in cloning a murine Fc fragment of IgG2a that works as a CXCR4 antagonist into the genome of the oncolytic vaccinia virus, was tested in ovarian CSCs and reduced the CSC population via a reduction in CXCL12 levels in ascites, and reduced recruitment of ECs, MDSCs and plasmacytoid DCs to the tumor. Moreover, it increased infiltration of CTLs due to the release of danger signals related with the viral infection and cellular lysis [223]. In addition, the use of siRNAs and the chemical antagonist CCX771 have been explored to target CXCR7 in gliomas, inhibiting the phosphorylation of ERK1/2 and obtaining similar results as when targeting CXCR4, although further research needs to be performed regarding CXCR7 inhibitors [213]. A summary of the different drugs used to target the CXCL12/CXCR4/CXC7 axes can be found in Table 2.

The effects of anti-CXCL12/CXCR4 therapies have been most extensively studied in hematological malignancies, in which monoclonal CXCR4 targeting antibodies such as Ulocuplumab, have been tested with varying results. Unfortunately, these monoclonal antibodies also showed cytotoxicity that affected normal hematopoiesis. While clinical trials (NCT01120457, NCT01359657) are being carried out to evaluate the therapeutic effects of these antibodies, most of these are still in phases I and II. CXCR4 antagonists generally show a secure profile as co-adjuvant therapies; however, they may also lead to side effects such as hyperleukocytosis or myeloid suppression [243,244]. Indeed, the impact of these therapeutic strategies on immune and stromal cells is also being studied with respect to their possible modulation of the TME.

PDAC is considered an immunologically “cold tumor”, because of the low infiltration of T effector cells. This is due to the high desmoplastic reaction and fibrotic stroma surrounding the tumor, which does not allow T effector cells to reach the cancer cells, making PDAC a non-suitable tumor for immunotherapy. Thus, strategies to eliminate stoma components, such as CAFs, have been developed, including the use of AMD3100 to deplete the CXCL12^+^ FAP^+^ CAF population, which results in a marked infiltration of T cells into the tumor and an increase in the effectiveness of anti-PDL1 therapy [240]. Moreover, the COMBAT trial (Phase IIa, NCT02826486) has shown that the combination of AMD3100 and pembrolizumab to treat PDAC resulted in higher response to chemotherapy, overall survival, and disease control rate with no significant side effects [245]. Accordingly, accumulation of MDSCs in osteosarcoma has also been shown to strongly contribute to anti-PDL1 therapy failure. AMD3100 co-treatment with anti-PD1 antibodies was reported to halt MDSCs recruitment and promoted an expansion of CTLs, which reduced tumor growth in murine models [226]. Moreover, the CXCL12/CXCR4 axis is thought to generate resistance to anti-VEGF therapy via recruiting monocytes and neutrophils in CRC, which fosters an immunosuppressive environment. AMD3100 treatment inhibited immune cell recruitment and led to an overall survival in an extended treatment approach in a mouse model of CRC [225]. Thus, blockade of the CXCL12/CXCR4/CXCR7 axes to treat different types of tumors appear to be promising approaches, which are currently being tested in different clinical trials. However, a more comprehensive understanding of how this inhibition affects all different cell populations involved in the CXCL12 crosstalk must and should be achieved.

As AMD3100 also inhibits CXCR7, further efforts are underway to identify more specific CXCR4 inhibitors. While a combined inhibition of CXCR4 and CXCR7 may be advantageous for cancer treatment by more completely inhibiting CXCL12 signaling, more highly specific inhibitors may significantly reduce the side effects linked to small molecule inhibitors. Epi-X4, a highly specific endogenous peptide with strong CXCR4 inhibitory capacity based on a cleavage product of human serum albumin has recently been identified [242]. While this peptide is already effective, short serum stability can be overcome by chemical modulation [241], making EPi-X4 derivatives interesting candidates for highly specific CXCR4 inhibitors that might be applied without serious side effects.

The existence of already approved drugs which could affect CXCL12 and/or its receptors was investigated by searching different data bases. Only one possible candidate was identified: AMD070 (Mavorixafor), an orphan drug used to treat WHIM syndrome (Warts, Hypogammaglobulinemia, Infections, and Myelokathexis), which is currently in a Phase III clinical trial (NCT03995108) [246]. AMD070 could have anticancer effects by inhibiting CXCR4, and it is being evaluated in Waldenström’s macroglobulinemia (WM), a rare form of non-Hodgkin’s lymphoma, in a Phase I clinical trial (NCT04274738).

## 6. Conclusions

Chemokines are well known for being main players in, and influencers of, the tumorigenic process. A prime example is CXCL12, which together with its receptors CXCR4 and CXCR7 guide tumor cell metastatic migration, recruit stromal cells (including immune cells), promote angio- and lymphogenesis and constitute a crucial part of the CSC niche in many tumor entities. However, there are some discrepancies regarding its role in specific tumor types as mentioned above, such as PDAC and CRC, which illustrates the complexity of CXCL12 signaling and its receptors in tumor maintenance and progression and highlights the need for further research to resolve these discrepancies. Nevertheless, the connection between the CXCL12/CXCR4/CXCR7 axes and CSCs across different tumor entities is undeniable and backed by numerous studies showing that these axes regulate the maintenance of normal and tumoral stem cells in a direct way, and that these axes are involved in the activation and/or reinforcement of CSC features, including self-renewal, chemoresistance, metastasis, and the expansion of the CSC compartment as described above. Not only do the CXCL12/CXCR4/CXCR7 axes affect CSCs directly, but they also influence CSCs indirectly by modulating the TME landscape towards a more CSC-supportive state. CXCL12 is critically involved in remodeling the immune landscape of the tumor towards a more immune suppressive environment by recruiting cell populations such as TAMs and MDSCs, which promote tumor progression. Moreover, tumor-derived CXCL12 gradients recruit MSCs that generally promote tumor and vessels growth, protect the CSC compartment from chemotherapy and radiation, and can even differentiate into CAFs to further amplify TME signals. As far as its role in the metastatic process is concerned, CXCL12 promotes the mobilization of cancer cells from the primary tumor to the metastatic niche; therefore, targeting the CXCL12/CXCR4/CXCR7 axes may significantly improve therapeutic outcomes in patients, although the precise mechanisms of how the CXCL12/CXCR4/CXCR7 axes participate in metastasis are still unclear and need to be further elucidated. Nevertheless, understanding that CSCs are a group of highly specialized and plastic cells that comprise the root of the tumor, are able to alter the outcome of many antitumoral therapies (radiation, and chemo- and immunotherapy) and are strongly influenced by chemokines such as CXCL12, will help when designing novel therapies aiming at disrupting CSCs and their niche. Indeed, intensive research in this field has led to the development of promising therapies targeting the CXCL12/CXCR4/CXCR7 axes, including chemical inhibitors, synthetic peptides, antibody-based treatments, and endogenous peptide inhibitors to treat tumors and target CSCs. However, multiple questions remain unsolved and thus, further exploration of the CXCL12/CXCR4/CXCR7 axes in tumor biology seems to be necessary for a comprehensive understanding of its impact on tumor progression, as well as on CSCs and the maintenance of their niche. Such insight would certainly facilitate the development of novel therapeutic strategies to achieve better clinical benefits.

## Figures and Tables

**Figure 1 cancers-13-00469-f001:**
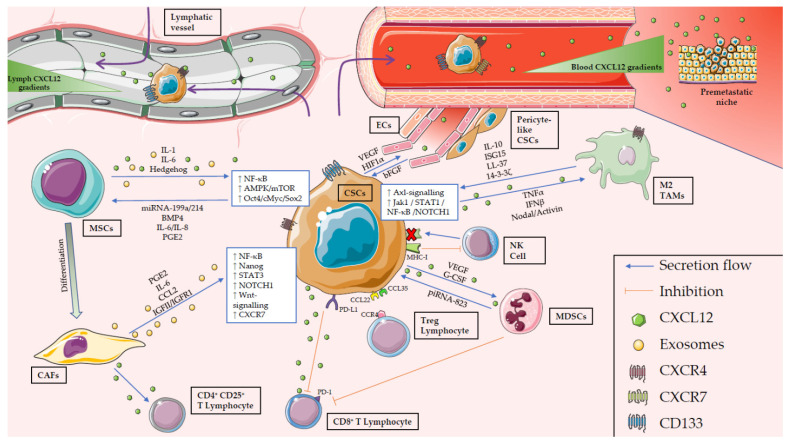
Schematic representation of the CSC niche and CXCL12 fluxes. CSCs are surrounded by different kinds of stromal cells which support their stemness and promote cell survival, proliferation, migration, immune evasion, and self-renewal. These processes are regulated by a myriad of cytokines, receptors, and extracellular vesicles. In particular, CXCL12 fluxes represent one of the key players in the CSC niche. CXCL12 is produced by MSCs and CAFs to induce stemness in tumor cells and recruit CD4^+^CD25^+^ T cells, respectively. It is also secreted by CSCs not only to recruit M2 TAMs and MDSCs, but also to inhibit possible interactions with CTLs. Importantly, CXCR12/CXCR4/CXCR7 axes strongly induce CSC migration from the primary tumor through a blood CXCL12 gradient that originates in the premetastatic niche. Finally, the interaction between stroma and endothelial cells with CSCs enhances their invasive and intravasive abilities.

**Figure 2 cancers-13-00469-f002:**
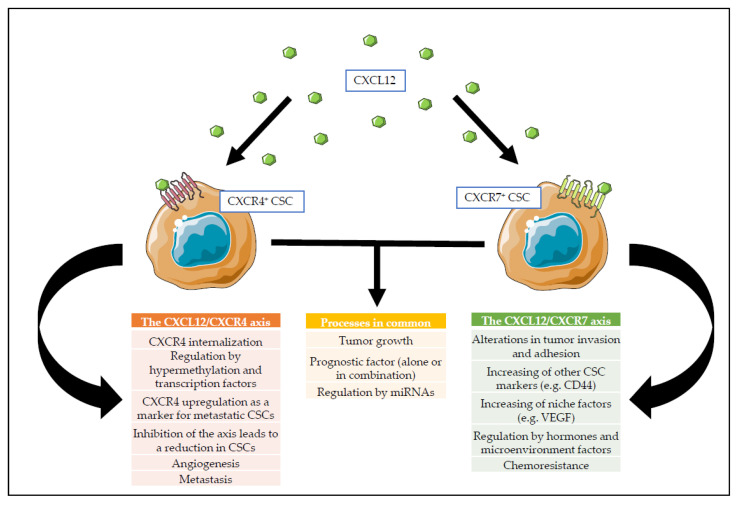
Comparison of the effects and processes initiated after the interaction between TME CXCL12 and CSCs, which express CXCR4 and/or CXCR7 on their cell surface.

**Table 1 cancers-13-00469-t001:** Effects of the CXCL12/CXCR4 axis in CSCs and their molecular mechanisms.

Tumoral Source of CSCs	CXCL12/CXCR4 Effects	Molecular Mechanisms
Breast cancer	Metastasis	Activation of RhoA [133].
	Adhesion	Interaction between RhoA and Rac1 [133].
	EMT	Activation of SOX2, OCT4 and NANOG [134]. Wnt/β-catenin pathway activation [135].
	Self-renewal	JNK/cJun overactivation [134].
Lung cancer	Self-renewal	mTOR and Akt signaling activation [136].
	Chemoresistance	Upregulation of CYP1B1 [137].
Squamous cell carcinomas	Migration	Increase podia formation [138].
	CSC expansion	Autocrine activation of PDGFRα [139].
Gastric cancer	Self-renewal	Wnt5a secretion by ILCs and RhoA activation in CSCs [140].
Esophageal cancer	Metastasis and stemness	Activation of ERK1/2 pathway [141].
Hepatocellular carcinoma	Stemness	Activation of Wnt/β-catenin pathway [142].
Pancreatic cancer	Metastasis	Loss of KLF10 and activation of c-Jun [143].
Glioma	Cell survival and tumor progression	Autocrine positive loop of CXCL12 [144,145].
Acute Myeloid Leukemia	Cell survival	Activation of anti-apoptotic pathways [146].

**Table 2 cancers-13-00469-t002:** Summary of the different drugs proposed to target the CXCL12/CXCR4/CXCR7 axes in cancer.

Drug	Molecule Inhibited	Tested in	Biological Effect
AMD3100 (Plerixafor)	CXCR4 and CXCR7	Medulloblastoma	Inhibition of cell migration, survival and proliferation [237]. Recruitment and expansion of CTLs, MDSCs halting [225,240].
		PDAC	
		Glioblastoma	
PRX177561	CXCR4	Glioblastoma	Reduction of tumor growth, M2 to M1 macrophage polarization and reduction of CSC traits [239].
TN14003	CXCR4	Breast cancer	Blocking metastasis [192].
Oncolytic virus	CXCR4	Ovarian cancer	CSC reduction and immune modulation [223].
EPI-X4	CXCR4	PDAC, leukemia	Inhibition of cell migration [241,242].

## Data Availability

Not applicable.

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
