# Peer review of "The CXCL12 Crossroads in Cancer Stem Cells and Their Niche"

_cancers, 2021, doi:10.3390/cancers13030469_

Round 1
Reviewer 1 Report
Comment:
In the [Cancers] Manuscript ID: cancers-1064327, authored by Juan Carlos López-Gil, Laura Martin-Hijano, Patrick C. Hermann *, Bruno Sainz, Jr. * attempted to exploit the gap of connection between the CXCL12/CXCR4/CXCR7 axes and CSCs across different tumor entities for the self-renewal, chemoresistance, metastasis of tumor progression, upon responding to treatment, chemotherapy, and radiation. The topic is of great interest.
They crawl freely, crossing the boundaries of different concepts. Some locations need a transition point to articulate where the problem ends and the solution begins - to make sure that the manuscript "flows" from one part to the next - that there are no gaps in the flow of ideas and that the manuscript ends up with a cohesive message and not a jumble of ideas/facts/data sets, so much to be specific, enabling the reader to tighten grip over such a complex problem. E.g., as critique #14 below: They should flush out a specific problem derived from the digestion of facts/data to the level of so thorough that predicament seems insurmountable, thereby offering the solution of their insight and perspective to the problem in the manuscript – that is what the reader looks forward to reading the literature review manuscript. Otherwise, the reader knew the facts/data, why bother the manuscript regurgitate it again?
Below specific comments and suggestions for the authors should be incorporated to improve its clarity, coherence, and logic flow.
Specific comments:
- The title “The CXCL12 Crossroads in Cancer Stem Cells and their Niche” did not reflect the content, as they didn’t articulate the point.
- Abstract: “Cancer stem cells (CSCs) are defined as a subpopulation of
“stem”-like cells within the tumor with self-renewal, metastatic, plastic
and chemoresistant capacities that allow them to maintain tumor growth,
escape standard anti-tumor therapies and drive subsequent repopulation of the
tumor.” This is not the classic definition of CSCs. Neither should be used for “the CSC niche, a compartment within the tumor microenvironment that includes a diverse group of cells focused on maintaining and supporting the CSC.”
- “In this review, we will provide the latest update on how CXCL12 and its receptors support the CSC niche and CSCs, as well as provide a summary of the up-to-date therapeutic options being currently tested to disrupt the CXCL12 axes in order to target, directly or indirectly, the CSC subpopulation.” They should rather have entailed what “summary” is about in Abstract – “up-to-date therapeutic options” – targets – than written down about the definition of CSCs, really wasting the reader’s time.
- Line 52: multipotent – They needed to elaborate on what is about here.
- Lines 71-72: They needed to define the tumor microenvironment (TME) first and then to move on to describe the CSCs evolution? Didn’t the literature define TME consists of CSCs niche? In fact, Line 132: “Along these lines, the CSC niche (as part of the TME), also plays an important role in driving …”
- Lines 75: “reinforcing the hypothesis that cancer stemness is more a state than an entity” – If the authors adopted this definition, they needed to use such along with the entire manuscript. And yet, they didn’t, causing the drifting logic.
- Lines 76-91: “Epithelial-to-mesenchymal transition (EMT)” came from Robert Weinberg’s lab, which denied the CSC concept. The way this MS confused the reader.
- Lines 102-122: “At the level of CSC markers, CD44, CD24, aldehyde dehydrogenase 1 (ALDH1), CD133 and CXCR4, the latter being a core part of this review, represent widely used markers to identify CSCs.” None of these is a definite CSC only. A balanced view should be implemented; otherwise, it is misleading.
- 1. The schematic diagram displays out of the proportion: e.g., extracellular vesicles look like CXCL12. What does that mean CSC with MHCI?
- Lines 189-204: “two different populations (anti- and pro-inflammatory), polarized” – “tumor-associated macrophages (TAMs)” – How did these differ from each other?
- Lines 790-815: “AMD3100 (Plerixafor)” – How did the side effect (cardiotoxic, Line 297) occur?
- Line 862: “maintenance and progression and highlights the need for further research.” Can they be specific to what they need?
- Line 867: “expansion of the CSC compartment” – How did that happen?
- Lines 867-873: “Not only do the CXCL12/CXCR4/CXCR7 axes affect CSCs directly, but they also influence CSCs indirectly by modulating the TME landscape towards a more CSC-supportive state. CXCL12 is critically involved in remodeling the immune landscape of the tumor towards a more immune-suppressive environment by recruiting cell populations such as TAMs and MDSCs, which promote tumor progression. Moreover, tumor-derived CXCL12 gradients recruit MSCs that generally promote tumor and vessel growth, protect the CSC compartment from chemotherapy and radiation, and can even differentiate into CAFs to amplify TME signals further.” Then needed to elaborate on “CSC-supportive state” – What’s the CSC state? What does the CSC-support mean? Keep CSC in CSC or become tumor cells? What did they relate to “tumor progression” or CSC state or vessels growth or chemotherapy or radiation? They crawl freely, crossing the boundaries of different concepts. Thus, some schemes should be used to center around these statements.
- Lines 874-877: “As far as its role in the metastatic process is concerned, CXCL12 promotes the mobilization of cancer cells from the primary tumor to the metastatic niche, although the precise mechanisms are still unclear and need to be further elucidated. Therefore, targeting CXCL12/CXCR4/CXCR7 may be necessary to improve therapeutic outcome in patients significantly.” It’s of confusion with such context.
Author Response
We thank Reviewer no.1 for his/her feedback, although we humbly disagree with many of the points raised. Nonetheless, we have addressed a large majority of the Reviewer’s comments and hope that the revised version of the manuscript is now satisfactory.
Specific comments:
- The title “The CXCL12 Crossroads in Cancer Stem Cells and their Niche” did not reflect the content, as they didn’t articulate the point.
Authors’ response: We kindly disagree with the Reviewer’s interpretation. We believe we have done a very good job review and highlighting this intersection, at the level of CXCL12, CSCs and their niche, and thus the title is appropriate. - Abstract: “Cancer stem cells (CSCs) are defined as a subpopulation of “stem”-like cells within the tumor with self-renewal, metastatic, plastic and chemoresistant capacities that allow them to maintain tumor growth, escape standard anti-tumor therapies and drive subsequent repopulation of the tumor.” This is not the classic definition of CSCs. Neither should be used for “the CSC niche, a compartment within the tumor microenvironment that includes a diverse group of cells focused on maintaining and supporting the CSC.”
Authors’ response: We kindly disagree with the Reviewer. The definitions used are commonly used in the field. In fact, we have published these definitions in many journals and feel that they are adequate and appropriate. - “In this review, we will provide the latest update on how CXCL12 and its receptors support the CSC niche and CSCs, as well as provide a summary of the up-to-date therapeutic options being currently tested to disrupt the CXCL12 axes in order to target, directly or indirectly, the CSC subpopulation.” They should rather have entailed what “summary” is about in Abstract – “up-to-date therapeutic options” – targets – than written down about the definition of CSCs, really wasting the reader’s time.
Authors’ response: We have edited the Abstract accordingly to stress the key points discussed in the Review. - Line 52: multipotent – They needed to elaborate on what is about here.
Authors’ response: To avoid confusion, we have removed the word “multi-potent” - Lines 71-72: They needed to define the tumor microenvironment (TME) first and then to move on to describe the CSCs evolution? Didn’t the literature define TME consists of CSCs niche? In fact, Line 132: “Along these lines, the CSC niche (as part of the TME), also plays an important role in driving …”
Authors’ response: While defined in the initial submission, we have stressed the definition of the TME. Lines 131-133 - Lines 75: “reinforcing the hypothesis that cancer stemness is more a state than an entity” – If the authors adopted this definition, they needed to use such along with the entire manuscript. And yet, they didn’t, causing the drifting logic.
Authors’ response: We kindly disagree with the Reviewer. We have used the concept of “state and plasticity” throughout the manuscript, have not drifted from our logic, and in fact use studies to reinforce this idea. See Lines: 228-231, 420-424, 598-606. - Lines 76-91: “Epithelial-to-mesenchymal transition (EMT)” came from Robert Weinberg’s lab, which denied the CSC concept. The way this MS confused the reader.
Authors’ response: We kindly disagree with the Reviewer. The first paper linking EMT with stemness was published by Mani et al, and from the Weinberg’s lab. We have cited this paper (Lines 86-90) to reinforce this concept, which is well accepted in the field. - Lines 102-122: “At the level of CSC markers, CD44, CD24, aldehyde dehydrogenase 1 (ALDH1), CD133 and CXCR4, the latter being a core part of this review, represent widely used markers to identify CSCs.” None of these is a definite CSC only. A balanced view should be implemented; otherwise, it is misleading.
Authors’ response: We kindly disagree with the Reviewer. The CSC field knows and accepts that no CSC marker, particularly those mentioned, are specific to CSCs. Nonetheless, we have reinforced that these markers are not CSC-specific: Line 105 - The schematic diagram displays out of the proportion: e.g., extracellular vesicles look like CXCL12. What does that mean CSC with MHCI?
Authors’ response: Schematic diagrams are routinely “not to scale. Likewise, CXCL12 is hexagonal and exosomes are circular; however, we do see where confusion could be found when blue exosomes are near green CXCL12 molecules. Thus, we have changed all the exosomes to yellow to contrast better with the green CXCL12 molecules.
Regarding MHCI, we have clarified this confusion both in the Diagram and in the corresponding legend. We hope these changes improve the Figure. - Lines 189-204: “two different populations (anti- and pro-inflammatory), polarized” – “tumor-associated macrophages (TAMs)” – How did these differ from each other?
Authors’ response: We kindly disagree with the Reviewer. There is no need to make a distinction between MSCs and TAMs. This is not the point of this section. We merely make a comparison.
However, we have edited this section to reduce confusion. Lines: 191-198 - Lines 790-815: “AMD3100 (Plerixafor)” – How did the side effect (cardiotoxic, Line 297) occur?
Authors’ response: The mechanism(s) underlying the cardiotoxic effects of AMD31000 are not important for this review. We have however, stressed that these effects were seen in HIV-infected patients. Line 814 - Line 862: “maintenance and progression and highlights the need for further research.” Can they be specific to what they need?
Authors’ response: We have specified what we mean as requested. Lines: 887-888 - Line 867: “expansion of the CSC compartment” – How did that happen?
Authors’ response: We refer the Reviewer to Lines 781-794, as one example of CSC compartment expansion. - Lines 867-873: “Not only do the CXCL12/CXCR4/CXCR7 axes affect CSCs directly, but they also influence CSCs indirectly by modulating the TME landscape towards a more CSC-supportive state. CXCL12 is critically involved in remodeling the immune landscape of the tumor towards a more immune-suppressive environment by recruiting cell populations such as TAMs and MDSCs, which promote tumor progression. Moreover, tumor-derived CXCL12 gradients recruit MSCs that generally promote tumor and vessel growth, protect the CSC compartment from chemotherapy and radiation, and can even differentiate into CAFs to amplify TME signals further.” Then needed to elaborate on “CSC-supportive state” – What’s the CSC state? What does the CSC-support mean? Keep CSC in CSC or become tumor cells? What did they relate to “tumor progression” or CSC state or vessels growth or chemotherapy or radiation? They crawl freely, crossing the boundaries of different concepts. Thus, some schemes should be used to center around these statements.
Authors’ response: We kindly disagree with the Reviewer. The Reviewer is asking to detail, in the “Conclusions” section all of these points, which have been extensively detailed and discussed in the main body of the paper. Doing so again here would be redundant and confusing. - Lines 874-877: “As far as its role in the metastatic process is concerned, CXCL12 promotes the mobilization of cancer cells from the primary tumor to the metastatic niche, although the precise mechanisms are still unclear and need to be further elucidated. Therefore, targeting CXCL12/CXCR4/CXCR7 may be necessary to improve therapeutic outcome in patients significantly.”
Authors’ response: We apologize that these sentences were confusing. We have edited them to improve clarity. Lines: 901-903
Reviewer 2 Report
This review by López-Gil et coll. focuses on the role of CXCL12/CXCR4/CXCR7 interactions for CSC maintenance, tumor growth and metastatic spread.
Overall, this review is comprehensive and meaningful.
Author Response
This review by López-Gil et coll. focuses on the role of CXCL12/CXCR4/CXCR7 interactions for CSC maintenance, tumor growth and metastatic spread. Overall, this review is comprehensive and meaningful.
Authors’ response: We thank Reviewer n.2 for his/her’s positive assessment of our manuscript.
Reviewer 3 Report
The work is well written and pleasant to read. However the authors could improve manuscprit by inserting a diagram illustrating the differences between the two ways indicated in:
paragraph 2.1 CXCL12 / CXCR4 axis and paragraph 2.2 CXCL12 / CXCR2 axis
Authors should discuss all cancers included in Table 1.
In the paragraph: "Therapeutic approaches to target the CXCL12 / CXCR4 / CXCR7 axes" the authors should divide the paraghraph in vitro effects and in vivo effects.
Moreover, the authors should clarify whether there are molecules already used in therapy which indirectly also act on the inhibition of CXCL12 / CXCR4 / CXCR7 axes
line 648 the authors wrote:
PDAC, we identified CXCR4 + CD133 + as an 649 exclusively metastatic subpopulation of PaCSCs
what cell line is it? Pancreatic? Authors should specify
Line 825: PDAC is considered an immunologically "cold tumor". Why?
Author Response
The work is well written and pleasant to read. However the authors could improve manuscript by inserting a diagram illustrating the differences between the two ways indicated in: paragraph 2.1 CXCL12 / CXCR4 axis and paragraph 2.2 CXCL12 / CXCR2 axis
Authors’ response: We thank the reviewer for his/her’s positive assessment of our manuscript. We agree with the Reviewer’s suggestion, and the manuscript now includes a new Figure (i.e. Fig. 2) showing how CXCL12 functions via each receptor separately.
Authors should discuss all cancers included in Table 1.
Authors’ response: We have reviewed the text and we are confident that all of the cancers mentioned in Table 1 are discussed in the main text.
In the paragraph: "Therapeutic approaches to target the CXCL12 / CXCR4 / CXCR7 axes" the authors should divide the paragraph in vitro effects and in vivo effects.
Authors’ response: While we appreciate the Reviewer’s suggestion and entertained the restructuring, in the end the suggested division created a confusing section with considerable repetition between parts and redundancy. Thus, we have left the section as is and hope the Reviewer understands our position.
Moreover, the authors should clarify whether there are molecules already used in therapy which indirectly also act on the inhibition of CXCL12 / CXCR4 / CXCR7 axes
Authors’ response: We thank the reviewer for this suggestion, and have included text to this effect (i.e. AMD070) in the manuscript. Lines: 870-876
line 648 the authors wrote: PDAC, we identified CXCR4 + CD133 + as an exclusively metastatic subpopulation of PaCSCs. What cell line is it? Pancreatic? Authors should specify
Authors’ response: We apologize for the lack of details, which are now included in the revised manuscript. Lines: 660-661
Line 825: PDAC is considered an immunologically "cold tumor". Why?
Authors’ response: We apologize for the lack of details, which are now included in the revised manuscript. Lines: 842-845
Round 2
Reviewer 1 Report
accepted